# Targeting Transcriptional Regulators Affecting Acarbose Biosynthesis in *Actinoplanes* sp. SE50/110 Using CRISPRi Silencing

**DOI:** 10.3390/microorganisms13010001

**Published:** 2024-12-24

**Authors:** Saskia Dymek, Lucas Jacob, Alfred Pühler, Jörn Kalinowski

**Affiliations:** 1Microbial Genomics and Biotechnology, Center for Biotechnology, Bielefeld University, 33615 Bielefeld, Germany; sdymek@cebitec.uni-bielefeld.de (S.D.); ljacob@cebitec.uni-bielefeld.de (L.J.); 2Senior Research Group in Genome Research of Industrial Microorganisms, Center for Biotechnology, Bielefeld University, 33615 Bielefeld, Germany; puehler@cebitec.uni-bielefeld.de

**Keywords:** *Actinoplanes*, acarbose, CRISPR interference, CRISPRi, library, transcriptional regulators

## Abstract

Acarbose, a pseudo-tetrasaccharide produced by *Actinoplanes* sp. SE50/110, is an α-glucosidase inhibitor and is used as a medication to treat type 2 diabetes. While the biosynthesis of acarbose has been elucidated, little is known about its regulation. Gene silencing using CRISPRi allows for the identification of potential regulators influencing acarbose formation. For this purpose, two types of CRISPRi vectors were established for application in *Actinoplanes* sp. SE50/110. The pCRISPomyces2i vector allows for reversible silencing, while the integrative pSETT4i vector provides a rapid screening approach for many targets due to its shorter conjugation time into *Actinoplanes* sp. These vectors were validated by silencing the known acarbose biosynthesis genes *acbB* and *acbV*, as well as their regulator, CadC. The reduction in product formation and the diminished relative transcript abundance of the respective genes served as evidence of successful silencing. The vectors were used to create a CRISPRi-based strain library, silencing 50 transcriptional regulators, to investigate their potential influence in acarbose biosynthesis. These transcriptional regulatory genes were selected from previous experiments involving protein–DNA interaction studies or due to their expression profiles. Eleven genes affecting the yield of acarbose were identified. The CRISPRi-mediated knockdown of seven of these genes significantly reduced acarbose biosynthesis, whereas the knockdown of four genes enhanced acarbose production.

## 1. Introduction

*Actinoplanes* strains are known for their ability to produce numerous secondary metabolites with antibacterial, antifungal, and antitumor properties [1,2,3]. The quest for novel active secondary metabolites has led to increased interest in rare actinomycetes, particularly *Actinoplanes* species [4]. Prominent examples of secondary metabolites with high clinical importance are the antibiotics actaplanin [5], ramoplanin [6], and teicoplanin [7] and also the pseudo-tetrasaccharide acarviosyl-1,4-maltose, known as acarbose [8,9]. Acarbose holds particular significance in pharmaceutical contexts for the treatment of type 2 diabetes. Its mechanism involves the inhibition of intestinal α-glucosidases, thereby reducing the absorption of monosaccharides in the human intestine and subsequently lowering blood sugar levels [10].

Regarding natural acarbose producers, research has focused on the Gram-positive actinobacterium *Actinoplanes* sp. SE50/110. Although this strain has been studied for over 20 years, its acarbose biosynthesis pathway has only recently been fully characterized [11]. Acarbose consists of two subunits: an amino-deoxyhexose and a C7-cyclitol. These subunits are synthesized through two biosynthetic pathways by AcbABVI and AcbCMOLNUJR, respectively. AcbS catalyzes the assembly of these subunits into acarbose. Through genomic sequencing [12], as well as transcriptomic [13,14], proteomic [15,16,17,18], and metabolomic [16] analyses, researchers have gained valuable insights into the complex life cycle and biosynthetic pathways of *Actinoplanes* sp. SE50/110. Despite this research, little is known about the regulation of acarbose biosynthesis. In contrast to other acarbose producers from the *Streptomyces* family [19,20], no transcriptional regulatory genes have been found within the acarbose biosynthetic gene cluster (*acb* cluster) [12]. Up to now, only the transcriptional regulators AcrC [21] and CadC [22] have been identified to have a regulatory influence on acarbose biosynthesis. AcrC is a repressor that binds directly to the intergenic region between *acbE* and *acbD*, while the binding site of CadC, a proposed transcriptional activator of *acb* genes, remains unidentified. Further regulators or binding sites are still unknown. Nevertheless, genomic, transcriptomic, proteomic, and metabolomic data have provided the foundation for identifying various genetic engineering targets for enhancing the production yields of acarbose and for understanding the regulation of acarbose biosynthesis and decreasing byproduct formation [21,23,24].

An essential part of characterizing and exploiting these capabilities is access to genetic engineering. Despite its high GC content of over 72%, slow growth, and complex life cycle, tools for genetic manipulation have been successfully developed in recent years for use in *Actinoplanes* sp. SE50/110 [3,12].

The transfer of DNA can be achieved through *Escherichia coli* (*E. coli*)–*Actinoplanes* sp. SE50/110 intergeneric conjugation [25]. Stable vector integration into the genome of *Actinoplanes* sp. SE50/110 is possible through the usage of ϕC31, ϕBT1, and VWB bacteriophage-based integrative vectors [25,26], while promoter screening leads to the creation of a promoter library, enabling heterologous expression to different degrees [27]. Deletion mutants can be generated using homologous recombination with *codA*, which serves as a counter-selection marker [28], or the ReDirect strategy [29]. Recently, the clustered regularly interspaced short palindromic repeats/(CRISPR)-associated endonuclease 9 (CRISPR/Cas9) system has mostly been employed for this purpose [30].

The vector pSET152, carrying the ϕC31 integrase gene, was further developed into pSETT4 to improve heterologous gene expression [26]. Its chromosomal integration site is located in ACSP50_6589, a gene annotated as a hypothetical pirin-homolog [25]. This integration site is homologous and is commonly used in actinobacteria, e.g., for manipulating the production of antibiotics in streptomycetes [31,32,33]. It has been proven to be a naturally occurring, unique, and stable integration site in *Actinoplanes* sp. SE50/110 [22].

Gene deletions in *Actinoplanes* sp. SE50/110 can be achieved using the CRISPR-Cas9 system on the pCRISPomyces2 vector [30,34]. This technique is based on the CRISPR system derived from *Streptococcus pyogenes*. The trans-activating CRISPR RNA (tracrRNA) forms an RNA duplex with the CRISPR RNA (crRNA), which is essential for the recruitment of Cas9 [35,36]. The Cas9 protein, comprising two nucleolytic domains, RuvC and HNH, creates specific double-stranded breaks [37]. The single guide (sgRNA) is created by developing a chimera from tracrRNA and crRNA [35]. The sgRNA is almost arbitrarily interchangeable, allowing any gene to serve as a target. The protospacer adjacent motif (PAM), located at the 3′ end of the sgRNA, is a crucial factor in this process. For *Streptococcus pyogenes*, this sequence is NGG [35,37,38]. Additionally, repair flanks are inserted in pCRISPomyces2, which enables targeted gene deletion via homology-directed repair [34].

Since the discovery of CRISPR-Cas9, it has been widely used to gain an understanding of the complex world of organisms. Its powerful potential is evident in its wide range of new applications, including the interaction of sgRNA and Cas9 variants [39,40]. An example of these new applications, which can be achieved with the modification of Cas9, is the targeted transcriptional silencing of genes, called CRISPR interference (CRISPRi) [41,42]. By exchanging two amino acids within the nucleolytic domains of RuvC and HNH (D10A and H841A), deadCas9 (dCas9) was created, which is no longer able to generate double-strand breaks in its target DNA [43]. The sgRNA dCas9 complex only binds to the sequence that is complementary to the sgRNA and prevents transcription by inhibiting the initiation or elongation steps of transcription [43]. This approach allows for a transient reduction in RNA from specific genes, whose deletion could otherwise be lethal.

However, the application in *Actinoplanes* sp. SE50/110 was limited to gene deletions until now. With the combination of dCas9 and pCRISPomyces2 or pSETT4, we have expanded the genetic engineering tool kit of *Actinoplanes* sp. SE50/110.

In a first step, the functionality of the systems in *Actinoplanes* sp. SE 50/110 was demonstrated. The genes of acarbose biosynthesis were used as an easily testable and quantifiable trait. The function of the *acb* genes in *Actinoplanes* sp. SE50/110 is known in most cases, as well as the transcription start sites [11,13]. The functionality of pCRISPomyces2i and pSETT4i was tested by repressing the known acarbose biosynthesis genes *acbB* and *acbV*, as well as that of the *acb* cluster activator *cadC*. The capability of pSETT4i as a fast screening method was proven in this work by investigating the influence of 50 transcriptional regulators on acarbose yield.

## 2. Materials and Methods

### 2.1. Bacterial Strains, Media, and Reagents

All cloning steps were conducted in *E. coli* DH5α [44]. *E. coli* ET12567/pUZ8002 was employed for conjugations [45]. Cultivation on solid medium was performed on LB agar plates at 37 °C, and in liquid medium using LB broth at 37 °C and 180 rpm. Antibiotics were added if necessary (25 μg mL^−1^ chloramphenicol, 50 μg mL^−1^ kanamycin, and apramycin 50 μg mL^−1^). Chemicals were delivered by Carl Roth (Karlsruhe, Germany) if not stated otherwise.

*Actinoplanes* sp. SE50/110 and resulting mutants were cultivated on soy flour mannitol (SFM) plates (20 g L^−1^ soy flour (Sobo Naturkost, Cologne, Germany), 20 g L^−1^ mannitol, 20 g L^−1^ agar, in tap water; adjusted to pH 8) for solid cultivations and in nutrient broth medium (NBS, 10 g L^−1^ glucose, 4 g L^−1^ peptone, 4 g L^−1^ yeast extract, 1 g L^−1^ MgSO_4_·7H_2_O, 2 g L^−1^ KH_2_PO_4_, 2 g L^−1^ K_2_HPO_4_) at 28 °C and 140 rpm for liquid cultivations. The liquid cultivations were carried out in baffled polycarbonate flasks (Corning, Corning, NY, USA). Antibiotics were added if necessary (50 μg mL^−1^ apramycin, 100 μg mL^−1^ fosfomycin).

LB medium was purchased from Thermo Fisher Scientific (Waltham, MA, USA), soya flour from Sobo Naturkost (Cologne, Germany), and agar from Becton, Dickinson & Company (Heidelberg, MN, USA). All other chemicals were sourced from VWR (Road Radnor, PA, USA), Sigma-Aldrich (St. Louis, MO, USA), and Merck (Darmstadt, Germany), if not stated otherwise.

All PCRs were conducted using the Phusion High-Fidelity PCR Master Mix (Thermo Fisher Scientific Inc., Waltham, MA, USA). The enzymes for the Golden Gate reaction were sourced from Thermo Fischer Scientific Inc (Waltham, MA, USA). Gibson assemblies were performed following the protocol outlined by [46] and the instructions provided by New England Biolabs (Ipswich, MA, USA).

Strains used in this study are listed in the Appendix A.

### 2.2. Construction of CRISPRi Vector Systems

The pCRISPomyces2 plasmid was employed as the starting point for this study. Single point mutations were introduced at the nucleolytic domains of RuvC (D10A) and HNH (H840A) via site-directed mutagenesis. Oligonucleotides used in this study are listed in the Appendix A. The resulting fragments were cloned via Gibson Assembly [46]. The resulting pCRISPomyces2i was verified by Sanger sequencing.

The sgRNAs were designed using the CRISPy-web tool [47], based on the genome sequence of *Actinoplanes* sp. SE50/110. The sgRNAs were designed as 20 bp long sequences unique to the genome, with the PAM being NGG. The sgRNA was designed to bind the non-template strand and, if possible, within the promoter or the open reading frame near the start codon. The sgRNAs were ordered as oligonucleotides with suitable overhangs (listed in Appendix A) and cloned into pCRISPomyces2i using the Golden Gate method and the type IIS restriction enzyme BbsI, as described previously [30]. The resulting plasmid was transformed into *E. coli* DH5α via heat shock.

To obtain pSETT4i, the silencing cassette, including dCas9 and the tracrRNA with the insertion site of the sgRNA, was amplified from pCRISPomyces2i and inserted into pSETT4 [26] via Gibson Assembly [46]. Since pSETT4 contains multiple BbsI sites in the integrase, the recognition sequences for insertion of the sgRNA were converted from BbsI to BsaI by site-directed mutagenesis. As the cleavage sites of both restriction enzymes remain identical, the designed sgRNA is compatible with both pCRISPomyces2i and pSETT4i.

### 2.3. Conjugation and Plasmid Curing in Actinoplanes

The conjugation of *Actinoplanes* sp. SE50/110 was conducted using *E. coli* ET12567/pUZ8002 as a conjugation host [45].

Initially, competent *Actinoplanes* cells were produced as previously described [27]. For this purpose, *Actinoplanes* sp. SE50/110 was grown in NBS, passaged in NBS after two days, and centrifuged (4 °C, 5000 rpm, 3 min) after a further two days. The cell pellet was washed in 10% (*w/v*) ice-cold sucrose, followed by centrifugation and washing in 15% (*w/v*) ice-cold glycerol. After another centrifugation step, the cells were dissolved in 15% (*w/v*) ice-cold glycerol and stored at −80 °C in 200 µL aliquots.

A 10 mL pre-culture of *E. coli* ET12567/pUZ8002 with the desired plasmid was grown overnight in LB with apramycin, kanamycin, and chloramphenicol at 37 °C and 180 rpm. The main culture of 50 mL was inoculated with an OD_600_ of 0.1 and grown in the same medium as the pre-culture until an OD_600_ of 0.6 was reached. The culture was centrifuged (4 °C, 5000 rpm, 3 min) and washed twice with 50 mL ice-cold NBS. The cell pellet was dissolved in 10 mL ice-cold NBS.

For each construct to be conjugated, five aliquots of competent *Actinoplanes* sp. SE50/110 cells were mixed with 10 mL of *E. coli* conjugation strain. Subsequently, 1 mL of the mixture was plated onto an SFM plate using a cotton swab for each construct. After 20 h, the plates were covered with 0.5 mg apramycin and 1 mg phosphomycin, dissolved in 1 mL of water. As soon as exconjugants had visibly grown (after one to eight weeks), they were transferred to SFM plates (50 µg mL^−1^ apramycin), and colony PCR was performed to confirm their genotype.

The curing of pCRISPomyces2i is possible due to the temperature-sensitive pSG5 origin of replication (*ori*) [48]. A single colony was picked from an SFM plate and grown in NBS for three days. The culture was passaged twice in NBS and cultivated at 37 °C and 140 rpm. The cell broth was diluted with 0.9% NaCl (1:2000, 1:5000), and 200 µL of the dilution was plated on SFM plates without antibiotics. The successful curing of the plasmid was confirmed by replica plating on SFM plates without and with apramycin (50 µg mL^−1^). Apramycin-sensitive colonies were transferred to NBS.

### 2.4. Cultivation of Actinoplanes

An *Actinoplanes* pre-culture was cultivated for 72 h at 28 °C and 140 rpm in NBS. Subsequently, 200 μL of the first pre-culture was transferred to 30 mL NBS. After 48 h, the second pre-culture was centrifuged (4 °C, 5000 rpm, 3 min) and washed in maltose minimal medium, and the cell wet weight was determined.

The main culture consisted of 50 mL maltose minimal medium as described elsewhere [15]. It was inoculated with 50 mg of washed cells and cultivated at 28 °C and 140 rpm until the stationary phase was reached. Samples for determining the cell dry weight (CDW), acarbose concentration, and RNA quantification were taken as described previously [21].

### 2.5. Acarbose Quantification

Acarbose present in the supernatant of *Actinoplanes* was quantified using high-performance liquid chromatography (HPLC) [27]. The supernatant was centrifuged (20,000× *g*, 2 min) to eliminate any residual particles. A total of 100 μL of the supernatant was mixed with 400 μL of methanol, vortexed, and centrifuged again (20,000× *g*, 2 min) to eliminate the resulting precipitate. The supernatant was then transferred to HPLC vials. The measurement was performed with the Agilent 1100 Series HPLC system (Agilent Technologies, Santa Clara, CA, USA) with a DAD detector. An injection of 40 μL of sample was separated at 40 °C using a Hypersil APS-2 column (125 × 4 mm, 3 μm, Thermo Fisher Scientific Inc., Waltham, MA, USA) with an isocratic gradient of 72% acetonitrile and 28% phosphate buffer (0.62 g L^−1^ KH_2_PO_4_, 0.38 g L^−1^ Na_2_HPO_4_ × H_2_O). Acarbose was quantified at a wavelength of 210 nm, employing a calibration curve generated by acarbose standards (Bayer AG, Wuppertal, Germany).

### 2.6. RNA Isolation and Quantification of mRNA Levels of Targeted Genes

The frozen cell pellet was dissolved in DNA/RNA Shield™ (Zymo Research, Irvine, CA, USA) and transferred to ZR BashingBead Lysis Tubes (0.1 and 0.5 mm) (Zymo Research, USA). Cell disruption was performed using a homogenizer (Precellys 24 homogenizer, Bertin Technologies, Montigny-le-Bretonneux, France) at 6500 rpm for 30 s. The homogenization process was repeated three times, with the cells stored on ice for 5 min in between. RNA isolation was conducted using the Quick-RNA Miniprep Kit (Zymo Research, USA) according to the manufacturer’s instructions. To confirm the elimination of residual DNA, PCR with primers binding to genomic DNA from *Actinoplanes* sp. SE50/110 was conducted. The quality and quantity of RNA were assessed using a NanoDrop 1000 spectrometer (Peqlab, Erlangen, Germany).

Reverse transcription quantitative PCR (RT-qPCR) was utilized to measure the relative mRNA levels of individual genes. Primers were designed to amplify intragenic regions spanning 75 to 150 base pairs. The RT-qPCR was carried out in 96-well lightcycler plates (Sarstedt, Nümbrecht, Germany) for measurements in a LightCycler 96 System (Roche, Mannheim, Germany). A total of 1 μL of template RNA, adjusted to 100 ng μL^−1^, was mixed with 19 μL master mix from the Luna Universal Probe One-Step RT-qPCR Kit (New England Biolabs, Ipswich, MA, USA), according to the manufacturer’s instructions. The program was implemented following the manufacturer’s instructions. At least three biological and two technical replicates were measured for each gene, and two negative controls with water instead of RNA were carried out.

To examine control measurements and analyze melting curves, the LightCycler 96 software (version 1.1.0.1320) was employed. The relative RNA quantity was normalized to the total RNA amount of 100 ng and calculated using 2^−ΔCq^. ΔCq was determined as the difference between the mean Cq values of the mutant and control strains.

## 3. Results

### 3.1. Design of Two Types of CRISPRi Vectors as a Basis for Silencing Experiments in Actinoplanes sp. SE50/110

As a basis for the usage of the CRISPRi system in *Actinoplanes* sp. SE50/110, two vector systems with distinct advantages were developed.

The basis of the first replicative pCRISPomyces2i silencing system is the pCRISPomyces2 plasmid, which is already used in *Actinoplanes* sp. SE0/110 for gene deletions [30]. The vector contains a codon-optimized *cas9* gene for high-GC actinobacteria. The temperature-sensitive replication origin (*ori*) of pSG5 is employed to cure the plasmid, and the apramycin resistance cassette is a selection marker. The freely selectable sgRNA can be inserted, e.g., via Golden Gate Assembly [49].

Firstly, dCas9 was created from the catalytically active Cas9 variant via site-directed mutagenesis. The known point mutations leading to *dCas9* are carried out in the domains RuvC (D10A) and HNH (H840A) [35].

The basis of the second, integrative pSETT4i silencing system is the plasmid pSETT4. The silencing cassette of pCRISPomyces2i, comprising *dCas9* and the tracrRNA with the insertion site of the sgRNA, was introduced into pSETT4 (Figure 1). Stable integration of the vector into the genome of *Actinoplanes* sp. SE50/110 is enabled by the bacteriophage integrase φC31. The integrase φC31 gene contains multiple BbsI recognition sequences. To ensure the efficient insertion of the sgRNA via Golden Gate Assembly, the BbsI recognition sequences were converted into BsaI recognition sequences by site-directed mutagenesis.

The protospacers for targeted silencing of distinct genes were designed using the CRISPy-web tool [47]. The genomic sequence of *Actinoplanes* sp. SE50/110 was used as the basis, ensuring that the 20 bp sgRNA sequence is unique in the genome and targets the non-template strand of the respective promoter region according to transcriptome sequencing [13]. If this was not possible, an sgRNA proximal to the 5′ region of the coding gene was selected.

### 3.2. Application of pCR_dCas9 for Reversible Silencing of the Genes acbB, acbV, and cadC 

Intrageneric conjugation of three silencing plasmids with sgRNA targeting *acbB*, *acbV,* and *cadC,* respectively, was successfully conducted, resulting in the *Actinoplanes* sp. SE50/100 strains pCR_dCas9_*acbB*^SI^, pCR_dCas9_*acbV*^SI^, pCR_dCas9_*cadC*^SI^. As a reference strain, *Actinoplanes* sp. SE50/110 carrying the vector pCRISPomyces2i without sgRNA was used. The strains were cultivated in maltose minimal medium for 192 h. Growth was recorded by cell dry weight, samples were taken to quantify acarbose production, and the yield was determined. RNA samples were taken during the early growth phase after 48 h. RT-qPCR was conducted, and the relative transcript amount was calculated using the 2 ^−ΔCq^ method.

The reference strain pCR_dCas9 achieved an acarbose yield of almost 0.05 g_acb_g_CDW_^−1^ (Figure 2A). The silenced strains pCR_dCas9_*acbB*^SI^, pCR_dCas9_*acbV*^SI^, and pCR_dCas9_*cadC*^SI^ showed a significantly lower yield than pCR_dCas9 (Figure 2A). The strain pCR_dCas9_*acbB*^SI^ showed a 50% lower acarbose yield of 0.024 g_acb_ g_CDW_^−1^, and pCR_dCas9_*acbV*^SI^ showed a reduction of 20% to 0.039 g_acb_ g_CDW_^−1^. Similarly, the acarbose yield of pCR_dCas9_*cad*C^SI^ decreased to 0.031 g_acb_ g_CDW_^−1^, a reduction of 38%.

RT-qPCR was conducted to determine whether the observed reduction in acarbose yield resulted from transcriptional silencing. The relative transcript levels of the silenced genes in all mutants decreased significantly by up to 17-fold compared to the reference strain (Figure 2B). Additionally, an expected regulatory effect on the genes *acbB* and *acbV* was observed in pCR_dCas9_*cadC*^SI^. Both genes were less transcribed in pCR_dCas9_*cadC*^SI^ compared to pCR_dCas9^SI^ since CadC is assumed to be a transcriptional activator of *acb* genes [22].

As pCRISPomyces2i carries the temperature-sensitive *ori* of plasmid pSG5, it is possible to cure the vector from the respective strain. This is achieved by cultivating the strains at 37 °C, plating the cells, and picking apramycin-sensitive colonies. The resulting strains pCR_dCas9_*acbB*^CU^, pCR_dCas9_*acbV*^CU^, and pCR_dCas9_*cadC*^CU^ were cultivated again, and the relative mRNA levels of the previously silenced genes were compared to those of the WT (Appendix A). No significant differences in transcription levels of the genes *acbB*, *acbV,* and *cadC* were detected in the cured strains.

Based on acarbose quantification and RT-qPCR analysis of transcript levels, the vector pCR_dCas9 was shown to be suitable for reversible transcriptional silencing in *Actinoplanes* sp. SE50/110.

### 3.3. Application of pSETT4i as a Rapid Screening Vector for Silencing of the Genes acbB, acbV, and cadC

pCRISPomyces2i is excellent as a reversible silencing system, but it has one drawback—the conjugation efficiency of this vector into *Actinoplanes* sp. SE50/110 is very low, and the generation of exconjugants can take up to two months. This limits the screening of many mutants within a limited timeframe.

To overcome this disadvantage, the integrative vector pSETT4 was complemented by the CRISPRi cassette of pCRISPomyces2i, creating the second silencing system pSETT4i. With this system, it takes less than a week for the exconjugates to be visible on agar plates. Both vectors, pCRISPomyces2i and pSETT4i, were designed to ensure the cloning compatibility of the sgRNA in both systems.

The integrative vector pSETT4i was tested for functionality with the same sgRNA used for pCRISPomyces2i before, creating the three strains pS_dCas9_*acbB*, pS_dCas9_*acbV*, and pS_dCas9_*cadC* in which the genes *acbB*, *acbV*, and *cadC* were silenced, respectively. These strains were compared with the reference strain pS_dCas9 without sgRNA. Cultivation and sampling were performed similarly to the pCRISPomyces2i system.

The yields of the strains pS_dCas9_*acbB*, pS_dCas9_*acbV*, and pS_dCas9_*cadC* were compared with the reference strain pS_dCas9. While the reference strain achieved an acarbose yield of 0.04 g_acb_ g_CDW_^−1^, the yields of the test strains were significantly lower by up to 23%, with values between 0.030 g_acb_ g_CDW_^−1^ and 0.034 g_acb_ g_CDW_^−1^ (Figure 3A).

In addition to the yield, the relative transcript levels of *cadC* and its target genes *acbB* and *acbV* in pS_dCas9_*cadC* were compared with those in the reference strain pS_dCas9. As before, for pCR_dCas9_*cadC*, relative transcript levels were significantly reduced, and transcriptional silencing was successfully demonstrated (Figure 3B).

Thus, pSETT4i is suitable as a CRISPRi system with the advantage of rapid conjugation compared to pCR_dCas9. This makes the system ideal for a high-throughput target screening.

### 3.4. Identification of Eleven Regulators Influencing Acarbose Biosynthesis Through Screening of CRISPRi Strain Library

Despite decades of research, only a few regulators influencing the *acb* cluster have been identified [21,22]. However, several putative regulators that influence acarbose biosynthesis have been suggested. The underlying evidence ranges from transcriptome and proteome analyses [13] to DNA affinity chromatography screening for transcriptional regulators binding to promoter sequences of the *acb* gene cluster (“regulator fishing”) [50]. The CRISPRi library target selection criteria are listed in the Appendix A.

A total of 50 regulators were screened with the newly established pSETT4i silencing vector to ascertain their influence on acarbose yield (Figure 4). The vector with the sgRNA targeting the respective gene was introduced into *Actinoplanes* sp. SE50/110, creating a CRISPRi strain library. Cultivation and sampling were conducted as previously described.

One of the criteria for target selection of the screening in this work was the preliminary experiment of regulator fishing using promoter sequences of the *acb* cluster. By this method, 28 putative regulators of the *acb* cluster were identified and were subjected to silencing by the pSETT4i system. Eight of these regulators showed an influence on the acarbose yield when silenced. The yield was found to be increased by up to 30% for silenced regulators ACSP50_4202, ACSP50_7823, and ACSP50_7982. The regulators ACSP50_8048, ACSP50_4228, ACSP50_1816, ACSP50_6463, and ACSP50_8173 were observed to reduce the yield by up to 88% when silenced. When silenced, the remaining 21 putative regulators identified by regulator fishing showed no significant influence on acarbose biosynthesis.

Transcriptome profiling analysis [13] identified ten regulators that cluster with the acb genes’ expression profiles. These were also selected for silencing. While ACSP50_6463 was previously mentioned and was also identified by regulator fishing and mentioned in a patent [51], silencing the nine remaining and synchronously transcribed regulators led to no significant influence on acarbose biosynthesis.

The regulator ACSP50_1877 was also identified through transcriptome profiling analysis. Unlike the previously mentioned synchronously transcribed regulators, ACSP50_*1877* showed a transcriptional pattern opposite to that of the *acb* cluster. Interestingly, its silencing had a strong effect and increased the acarbose yield by approximately 30%.

In addition to regulator fishing and transcriptome profiling, homologs of the known global regulator CadC, which among others also influences acarbose biosynthesis, were selected as another category for silencing targets. One of these paralogs, ACSP50_4202, with an identity value of 38.27% according to BlastP, was identified through both regulator fishing and homology comparison. Although the regulators ACSP50_1755 and CadC exhibited the highest similarity of 56.70%, no difference in acarbose yield was observed compared to the reference strain when ACSP50_*1755* was silenced. However, the acarbose yield in the strains in which the regulators ACSP50_4574 (46.54% identity) and ACSP50_519 1 (38.89% identity) are silenced decreased by 20% and 47%, respectively.

Out of the 50 regulators screened, a total of 11 were found to exert a significant influence on the acarbose yield. Seven of these targets resulted in a decrease in yield, while four led to an increase.

## 4. Discussion

This work established two types of CRISPRi vectors for use in *Actinoplanes* sp. SE50/110 to extend the range of methods available for this organism, which is generally difficult to access by genetic engineering methods.

Besides other tools for functional gene characterization, such as heterologous or inducible expression and antisense RNA silencing [52,53], CRISPRi offers an advantageous extension for these characterizations due to its simplicity and broad application possibilities. Only dCas9 and an sgRNA targeting the gene of interest are required. These two modules also enable the tuning of knockdown levels, multiplexing, and the targeting of essential genes [54].

As pCRISPomyces2i enables reversible silencing, in contrast to a deletion system, it is of particular interest for targets whose deletion would be lethal either in principle or under certain growth conditions [55]. In this context, it is also possible to easily restore the wild type genetically and to unequivocally attribute the effects observed in the mutant to the silencing of the respective gene.

The system was evaluated by examining the acarbose biosynthesis enzymes AcbB and AcbV, along with their known transcriptional activator CadC. Both AcbB and AcbV are directly involved in the biosynthesis of acarbose. They catalyze the conversion of dTDP-D-glucose to dTDP-4-keto-6-deoxy-D-glucose, which is subsequently converted to dTDP-4-amino-4,6-dideoxy-D-glucose [56].

CadC is a global transcriptional activator in *Actinoplanes* sp. SE50/110, and its deletion by the ReDirect system had been observed to result in a reduction in transcription of the *acb* genes and a corresponding decrease in acarbose yield [22]. The silencing of the corresponding genes significantly impacts the yield of acarbose. In addition to determining the product yield, the target genes’ relative transcript levels were analyzed. A notable decline was observed in comparison to the reference strain. Therefore, the reduction in product yield observed in the strains pCR_dCas9_*acbB*, -*acbV,* and -*cadC* can be attributed to diminished transcription of the respective genes, indicating successful silencing. Moreover, a regulatory effect of CadC was demonstrated. By silencing *cadC*, the transcription of *acbB* and *acbV* was diminished, likely leading to decreased acarbose production. The curing of the plasmid was also successful, meaning that the wild-type situation could be restored with no significant differences in acarbose yield and in relative expression of respective genes compared to the wild type.

While the pCRISPomyces2i has many potential applications, one significant drawback for its use in *Actinoplanes* sp. SE50/110 is the long conjugation time, extending up to two months. This renders the screening of numerous targets exceedingly challenging. Because of this, pSETT4i was developed with a reliable conjugation time of less than one week.

Due to the single genomic integration of the vector, the copy number is much lower compared to pCRISPomyces2i, which has a copy number of approximately 40 per chromosome in *Streptomyces* [48]. To address this issue, the slightly stronger *tipA* promoter is employed instead of the *rpsL* promoter [27].

The functionality of pSETT4i was tested in the same way as before for pCRISPomyces2i. Again, a significant reduction in acarbose yield in the strains with silenced *acbB*, *acbV,* and *cadC* was observed compared to the reference strain. Furthermore, a decrease in the relative transcription rate of *cadC* and its regulated genes was identified. Thus, pSETT4i is an excellent CRISPRi system to generate a CRISPRi strain library in a short time, e.g., for screening initiatives.

CRISPRi screenings have been applied to enhance the biosynthesis of decaprenoxanthin in *Corynebacterium glutamicum* [57], to stimulate antibiotic production in *Streptomyces* [58], or to increase α-amylase yield in *Bacillus subtilis* [59]. Furthermore, the system is also employed to suppress the formation of undesirable byproducts [60,61].

In this study, the influence of several regulators on acarbose biosynthesis was investigated. In contrast to the majority of secondary metabolite clusters, *Actinoplanes* sp. SE50/110 lacks a regulator within the *acb* cluster [62]. Furthermore, it should be noted that acarbose is not a typical secondary metabolite, as it is produced exclusively during active growth [13]. Thus, a more complex integration into a regulatory context can be assumed. It has been demonstrated that components of sigma factor regulation exert an (indirect) influence on acarbose synthesis [63]. Accordingly, a multiple “guilt by association” strategy was employed to identify potential targets. The initial criterion is the physical binding of the regulator protein to putative promoter regions, a process known as “regulator fishing”. This approach can be applied without further knowledge and targets regulators that have a direct influence by binding to the promoters of biosynthesis genes. Another criterion was the similarity of the transcriptional profiles (clustering) to those of the target genes. In addition to direct regulatory influences, indirect influences are also targeted in this approach. Thirdly, the strategy encompasses gene families, wherein the hypothesis is that regulators within a family may exert control over similar target genes.

As mentioned, one of the criteria employed in the selection of the targets within the library was the synchronous transcription with the *acb* gene cluster. This encompasses ACSP50_*6463*, which is transcribed in parallel with the monocistronic *acb* genes [13], and whose silencing resulted in a notable reduction of 88% in yield. Of particular note is the anti-parallel transcribed regulator ACSP50_*1877* [13], which has been observed to increase acarbose yield when silenced. It may be postulated that it acts as a repressor of the *acb* cluster. While overexpression of ACSP50_*1816* has been observed to increase acarbose production before [52], its repression has been found to reduce it by 25% in this work. RNA-Seq data indicate that this regulator has a consistently high transcript level in *Actinoplanes* sp. SE50/110, suggesting it may act as a global regulator [14].

It has been observed that certain regulatory proteins in *Actinoplanes* sp. SE50/110 exhibit homology to the global regulator CadC, which influences acarbose biosynthesis. Some of these have been subjected to screening in the library and were found to influence the yield of acarbose. ACSP50_4202 shows an increased yield, while the yields of ACSP50_4574 and ACP50_5191 decrease when the respective genes are silenced. These regulators are members of the ArsR family, a group of repressors whose binding to DNA is dependent on the presence of metal ions [64]. Members of this family are involved in different cellular processes. In *Streptomyces roseosporus* and *Streptomyces verticillus*, for instance, they regulate the biosynthesis of the antibiotics daptomycin and bleomycin, respectively [65,66]. In addition, ArsR family regulators have been demonstrated to exert an influence on the processes of cell death and sporulation. In Streptomyces, links between sporulation and the production of secondary metabolites have already been demonstrated [67,68]. For example, the deletion of *rshA* in *S. coelicolor* leads to increased sporulation and prevents actinorhodin production [69]. It is therefore of particular interest that these regulators be further analyzed, not only in terms of their direct regulation of the *acb* cluster, but also regarding their role in the regulation of sporulation.

The regulator ACSP50_8048, whose silencing leads to a reduction in acarbose yield, may also act globally in *Actinoplanes* sp. SE50/110. It is one of the most highly transcribed genes in *Actinoplanes* and exhibits homology to the CarD family of transcription factors [70]. In *Myxococcus xanthus*, the CarD protein has been demonstrated to regulate light-induced carotenogenesis [71] Evidence suggests that carotenoid production influences acarbose yield in *Actinoplanes* sp. SE50/110. It was previously assumed that this was due to metabolic stress caused by carotenogenesis [24]. However, it has been demonstrated that a carotenoid biosynthesis cluster is synchronously transcribed with the *acb* cluster [13]. Thus, a common regulation or a regulation dependent on different metabolites is possible.

*Actinoplanes* sp. SE50/110 lacks pathway-specific regulators encoded in the *acb* cluster. In this context, a co-regulation, or at least a strong dependence, with other secondary metabolites should be considered. There is also sufficient evidence of a link between sporulation and acarbose biosynthesis [63]. The CRISPRi library was employed to identify both putative repressors and activators of the *acb* cluster, thereby narrowing the target list. The subsequent phase of the investigation will entail an examination on a molecular level, as it is not yet possible to ascertain whether the identified regulators exert a direct influence on the promoters of the *acb* genes. The creation of deletion mutants would be a common approach for investigating on–off effects [21,55,63]. CRISPRi, however, can facilitate a system-level comprehension of an organism, enabling the evaluation of responses to gradual rather than binary alterations resulting from gene deletion. Yet, combining different genetic engineering tools, including overexpression, silencing, and deletion, with omics technologies can provide a comprehensive insight into the modes of action of the regulators investigated.

## 5. Conclusions

This work established two CRISPRi vectors for use in *Actinoplanes* sp. SE50/110. The reversible silencing system of pCRISPomyces2i is particularly suitable for targeting genes whose deletion would be lethal. With pSETT4i, a rapid screening of CRISPRi libraries is enabled. Here, the system was used to analyze 50 regulators regarding their influence on acarbose production, of which 11 were identified as targets for further investigation. Seven of the silenced regulators were found to result in a reduction in acarbose yield. It can be speculated that overexpression of these regulators is beneficial for acarbose production. For the remaining four regulators whose silencing increases acarbose yield, the potential for further improvement in yield may be achieved through the deletion of the respective gene.

## Figures and Tables

**Figure 1 microorganisms-13-00001-f001:**
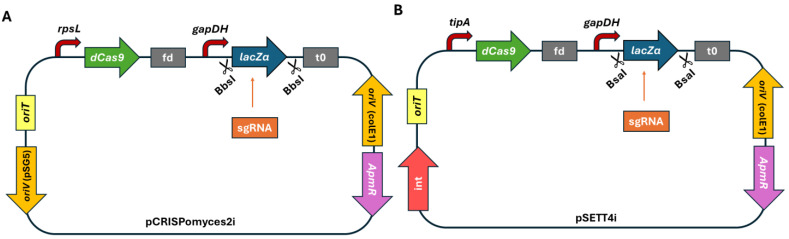
Design of the CRISPRi expression plasmids pCRISPomyces2i (**A**) and pSETT4i (**B**). *oriT*: origin of transfer; *rpsL*: promoter from *Xylanimonas cellulosilytica*; *dcas9*: modified from the *cas9* (Csn1) endonuclease from the *Streptococcus pyogenes* Type II CRISPR/Cas system; fd: terminator from bacteriophage fd; *gapDH*: promoter from *Eggerthella lenta*; BbsI: cloning site; *lacZα*: fragment from β-galactosidase from *E. coli*; sgRNA: individually designed; t_0_: terminator from phage lambda; *oriV* (colE1): high-copy-number origin of replication in *E. coli*; *apmR*: apramycin resistance from *Salmonella typhimurium*; *oriV* (pSG5): temperature sensitive replication initiator protein from the *Streptomyces ghanaensis* plasmid pSG5; int: integrase from phage φC31; *tipA*: promoter from *S. lividans*; BsaI: cloning site.

**Figure 2 microorganisms-13-00001-f002:**
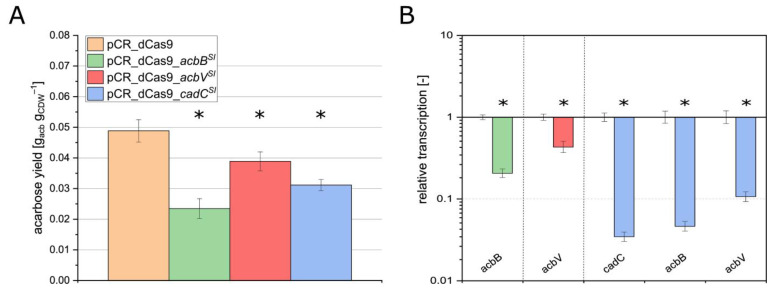
Acarbose yield of strains carrying pCRISPomyces2i silencing plasmids targeting *acbB*, *acbV,* and *cadC,* respectively, as well as the reference strain carrying pCRISPomyces2i without sgRNA (**A**) and relative transcript amount of *acbB* in strain pCR_dCas9_*acbB*^SI^, *acbV* in strain pCR_dCas9_*acbV*^SI^, and *cadC*, *acbB*, and *acbV* in strain pCR_dCas9_*cadC*^SI^ compared to the reference strain (**B**). Colors are used to label the different strains. Mean values and standard deviations of three biological replicates are presented. The significance (*p*-value < 0.05, labeled by *) was determined by Student’s *t*-test (unpaired, two-sided).

**Figure 3 microorganisms-13-00001-f003:**
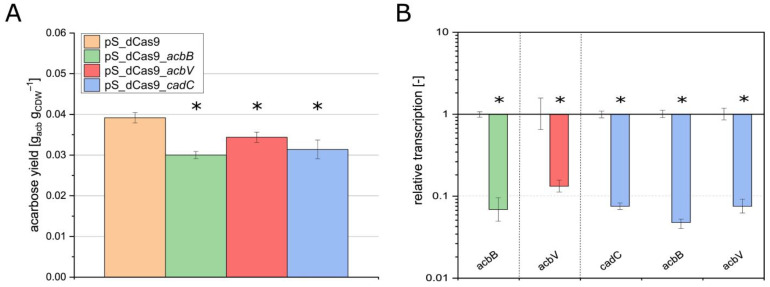
Acarbose yield of strains carrying pSETT4i silencing plasmids targeting *acbB*, *acbV,* and *cadC,* respectively, as well as the reference strain carrying pSETT4i without sgRNA (**A**) and relative transcript amount of *acbB* in strain pS_dCas9_*acbB*, *acbV* in strain pCS_dCas9_*acbV*, and *cadC*, *acbB*, and *acbV* in strain pS_dCas9_*cadC* compared to the reference strain (**B**). Colors are used to label the different strains. Mean values and standard deviations of three biological replicates are presented. The significance (*p*-value < 0.05, labeled by *) was determined by Student’s *t*-test (unpaired, two-sided).

**Figure 4 microorganisms-13-00001-f004:**
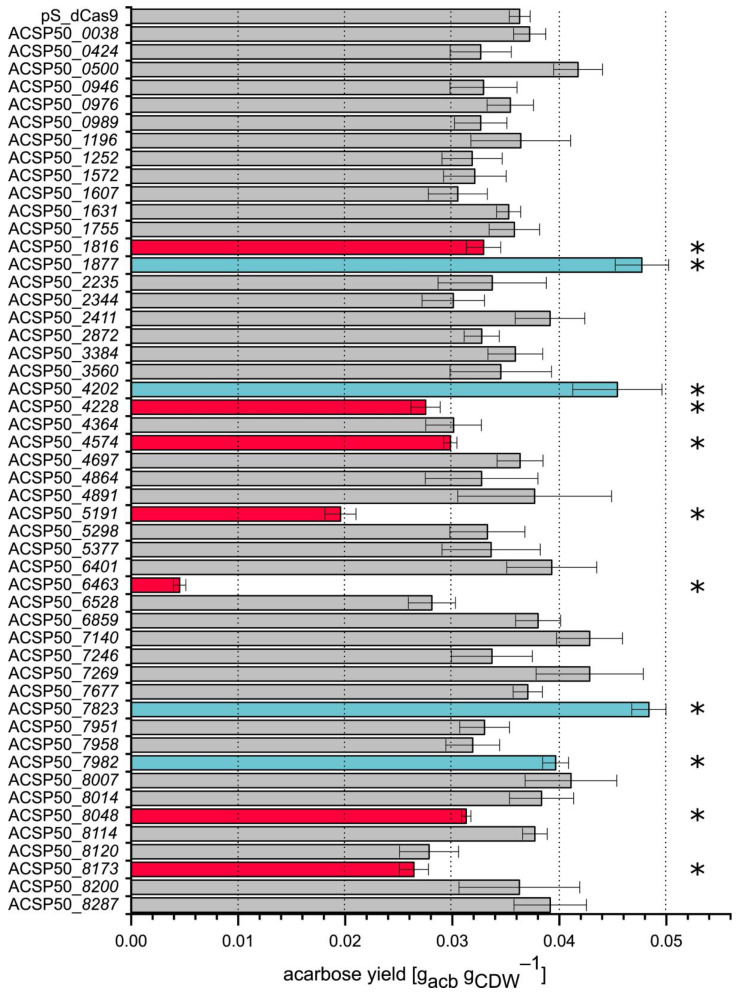
Effect of CRISPRi-mediated repression of transcriptional regulators associated with acarbose biosynthesis on acarbose yield compared to the reference strain pS_dCas9. The blue columns indicate a significantly higher yield compared to pS_dCas9, while the red columns indicate a significantly lower yield. Mean values and standard deviations of three biological replicates are shown. The significance (*p*-value < 0.05, labeled by *) was determined by Student’s *t*-test (unpaired, two-sided).

## Data Availability

The original contributions presented in this study are included in the article/Appendix A; further inquiries can be directed to the corresponding author.

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
