# Peer review of "Targeting Transcriptional Regulators Affecting Acarbose Biosynthesis in Actinoplanes sp. SE50/110 Using CRISPRi Silencing"

_microorganisms, 2024, doi:10.3390/microorganisms13010001_

Round 1

Reviewer 1 Report

Comments and Suggestions for Authors

This manuscript (MS) deals with development of vectors for gene silencing via CRISPR/dCas9-based interference system in Actinoplanes sp. SE50/110 and their application to screening transcriptional regulators involved in acarbose production. The operation of both vectors (pCRISPomyces2i and pSETT4i) was verified through silencing of three target genes (acbB, acbV, and cadC). The pSETT4i vector was also operated well in screening for regulators involved in acarbose biosynthesis. This work is expected to contribute to the research on Actinoplanes, where genetic engineering tools are relatively limited. However, some problems should be solved for possible publication. 

Major comments

-          L. 269-277: It is recommended moving this explanation to Introduction section.

-          Fig. 3(B): Results for relative transcription of pS_dCas9_acbB and pS_dCas9_acbV strains are missing. Why?

-          Table S4: Target gene selection criteria seems important in the flow of this study. However, the existence of Table S4 is not indicated anywhere in the main text. It would be good to indicate Table S4 in the text to improve the reader's understanding.

Minor comments

-          L. 45: metabolomic

-          L. 66: codA – italic

-          L. 68: (CRISPR)

-          L. 138: enzyme

-          All restriction enzymes should be italicized.

-          L. 174-175: apramycin was repeated.

-          L. 230: cas9 gene

-          Fig. 1: Genes in Fig. 1 should also be italicized.

-          L. 261 and 318: It is recommended to remove ‘Successful’.

-          L. 285: RT-qPCR

Author Response

see attached document

Reviewer 2 Report

Comments and Suggestions for Authors

1.      Line 141: What was amplified from pCRISPomyces2i?

2.      Line 164: What is an SFM plate? Please expand all abbreviations when they first appear in manuscript.

3.      Line 203: Please be consistent with grammatical tense. Most of the manuscript until now is written in past tense. Why then 'RNA isolation is conducted...'?

4.      Line 263 - 267: If there is a separate discussion section, there is no need to write this. Please only discuss results in 'Result' section.

5.      Line 496: Please provide reference for the assertion that 'ACSP50_8048 exhibits homology to the CarD family of transcription factors'.

6.      Why are supplementary tables (S2 to S4) not cited in main text?

Author Response

see attached document

Reviewer 3 Report

Comments and Suggestions for Authors

The manuscript presents a well-structured study on using CRISPRi systems to investigate transcriptional regulators affecting acarbose biosynthesis in Actinoplanes sp. SE50/110. The research addresses a significant gap in understanding acarbose production regulation and contributes valuable insights into genetic engineering for secondary metabolite production.

  Lines 11-12: The phrase "the biosynthesis of acarbose has been elucidated" is too general. Include more details about the biosynthetic pathway and the key enzymes or genes involved to provide readers with a clearer understanding. For instance, briefly outline the primary steps and mention pivotal enzymes like AcbB or AcbV.

  Lines 16-17: The statement "the pSETT4i vector providing a more efficient introduction into Actinoplanes sp." should be elaborated. Highlight specific advantages over pCRISPomyces2i, such as shorter conjugation time, improved integration efficiency, or enhanced stability. Support this claim with quantitative data if available.

  Lines 29-30: The introduction lists examples of secondary metabolites, but their relevance to acarbose research is unclear. Add a sentence to connect these metabolites' clinical importance with acarbose production, enhancing the transition between general context and the specific study.

  Lines 48-50: AcrC and CadC are introduced as repressors, but their mechanisms remain vague. Explain how they influence the acarbose biosynthetic pathway, such as their binding sites on gene promoters and the specific transcriptional activities they regulate.

  Lines 55-56: The mention of "omics data" is insightful but lacks specificity. Specify the types of omics analyses (e.g., genomics, transcriptomics, proteomics) that informed target selection and briefly explain their importance in identifying regulatory genes for engineering acarbose biosynthesis.

  Lines 70-73: Provide context for why the ACSP50_6589 gene was selected as the integration site for pSETT4. Explain its functional relevance or why it is a stable and commonly used site for integration in actinobacteria.

  Lines 94-95: Expand on why specific amino acid mutations (D10A and H840A) were chosen to generate dCas9. Emphasize how these changes ensure CRISPRi functions as a transcriptional inhibitor without causing DNA cleavage, reinforcing the method’s reliability.

  Lines 148-150: The preparation of competent Actinoplanes cells could be more detailed. Explain the rationale for the chosen conditions, such as why NBS media, centrifugation, and glycerol washing are ideal for preparing and preserving cell competency.

  Lines 193-196: The description of the chromatography setup for acarbose quantification is clear but could benefit from justification for the isocratic gradient choice. Explain how this setup efficiently separates acarbose from other metabolites, adding depth to the methodology.

  Lines 333-335: The comparative analysis between pCRISPomyces2i and pSETT4i is insightful but needs supporting quantitative data, such as conjugation efficiency, transformation success rates, or time required for colony formation. This would make the claim about pSETT4i’s advantages more compelling.

Author Response

see attached document

Round 2

Reviewer 3 Report

Comments and Suggestions for Authors

I'm very satisfied with the author's revisions, and I believe the paper is ready for acceptance.

Author Response

Dear Reviewer,

thank you very much for agreeing with our revision.

Best regards

Jörn Kalinowski (on behalf of all coauthors)